# Anxiety, Stress and the Resilience of University Students during the First Wave of the COVID-19 Pandemic

**DOI:** 10.3390/healthcare10122573

**Published:** 2022-12-19

**Authors:** Chrysoula Dafogianni, Despoina Pappa, Polyxeni Mangoulia, Freideriki Eleni Kourti, Ioannis Koutelekos, Evangelos Dousis, Nikoletta Margari, Eftychia Ferentinou, Areti Stavropoulou, Georgia Gerogianni, Evangelos Fradelos, Afroditi Zartaloudi

**Affiliations:** 1Department of Nursing, University of West Attica, 12243 Athens, Greece; 2Department of Nursing, National and Kapodistrian University of Athens, 10679 Athens, Greece; 3School of Medicine, National and Kapodistrian University of Athens, 10679 Athens, Greece; 4Faculty of Nursing, School of Health Sciences, University of Thessaly, 41500 Larissa, Greece

**Keywords:** psychological status, mental health, coping, university, students, COVID-19 pandemic

## Abstract

The COVID-19 pandemic had an impact on everyone’s daily lives with short-term or long-term consequences. Among the affected population, university students were studied by researchers specifically due to the total change to their educational way of learning and the courses they attended. The present study aimed to assess the psychological difficulties experienced by the university students of Greece during the first wave of the outbreak. Methods: 288 university nursing students completed an electronic questionnaire after consent. The sample included students from all years of study. The questionnaire included demographic data and questions about mental health status, resilience level, coping strategies, positive and negative emotions and an optimism assessment. Results: Depression (44.8%), anxiety (36.8%) and stress (40.3%) were experienced by the students. Females had significantly greater anxiety and stress signs compared to males (*p* < 0.001). The resilience score was significantly greater in males, as it was for the Positive Affect Score. Students in the fourth year of study used significantly more active/positive coping strategies than students in the first (*p* = 0.016) or second year of study (*p* = 0.005). Conclusion: Several students experienced serious mental disorders during the first period of the COVID-19 outbreak. Variables such as gender, year of study, age, positive and negative affect score, life orientation test score and coping strategies were identified as factors contributing to this situation. Special attention must be paid to female students as they mentioned negative emotions more frequently than males. Further research on the academic population could be beneficial to university administrators.

## 1. Introduction

University students’ daily lives were ultimately affected by the COVID-19 pandemic, which was perceived as an extremely stressful condition. Students’ academic studies were abruptly and suddenly disrupted due to social distancing preventive measurements leading to the disruption of face-to-face teaching and new virtual/online forms of learning at universities [1]. Academic institutions were closed down and distance learning was adopted as stay-at-home orders imposed quarantine, limited social relationships and isolation [2].

The adoption of the aforementioned measures led to positive outcomes related to public health protection, but it is necessary to emphasize that a negative effect was also noted in relation to the physical, psychological and social variables related to students’ health status. As students were forced to stay home, intimate interactions with classmates and peers were limited and, as a result, their mental health was affected. The increase in students’ perceived stress had a negative effect on their quality of life [3,4]. Students’ plans completely changed during the lockdown period and they experienced a breakdown of their relationships with classmates and mentors. Loss of the students’ access to their close relationships and friends, members of their campus community, as well as the structure and pace of the academic year made them more vulnerable. The prevailing unpredictable and uncontrollable conditions of the pandemic crisis contributed significantly to the increase in perceived stress and loss of people’s lives [5].

According to studies, Aylie et al. [6] concluded that there were higher levels of stress in 32.5% of the university students, while Kaparounaki et al. [7] measured higher levels of anxiety, depression and suicidal thoughts (with percentages of 42.5%, 74.3% and 63.3%, respectively) in a sample of Greek students during the long period of the COVID-19 pandemic. Students are vulnerable to a further aggravation of these feelings due to social distance, uncertainty and abrupt transitions [8]. Academic performance and students’ psychological well-being and quality of life can be affected by ongoing stress [9]. Coping (e.g., having a positive or avoidant attitude) is of great importance in order to reduce, minimize or tolerate as well as prevent stress [10]. Resilience and optimism have been reported to affect perceived stress and facilitate students’ coping strategies with stressful events in their university life [11]. 

This study aims to evaluate the perceived anxiety, stress, depression, the way of coping as well as the influence of optimism and resilience among Greek student participants during the COVID-19 outbreak.

## 2. Materials and Methods

### 2.1. Study Population

A cross-sectional study was conducted in five Greek Universities on the stress of students during the first period of the COVID-19 pandemic. Specifically, the research tool was distributed from June 2020 to August 2020. The sample consisted of nursing students. The specific study was carried out by completing a questionnaire in electronic form by students of the academic year 2019–2020, which was posted on the universities’ websites. The purpose of the online survey was to avoid direct contact, but also to encourage a large percentage of students to participate. The procedure was processed after approval by the Ethics Committee of the University of West Attica (Approval number: 52651). Participation was anonymous and voluntary, and the students filled out a consent form, declaring their agreement to participate in this study. 

### 2.2. Study Instruments

The first data recorded by the participants were demographic characteristics related to age, sex, place of residence and educational level. Following this part, five validated questionnaires assessing mental health status were completed by the study population.

Depression, anxiety and stress scale (DASS-21) [12], is a 21-item questionnaire designed to record, through a 4-point Likert scale, the severity of stress, depression and anxiety of the participants. The stress scale assesses irritability, hyperarousal, impatience and difficulty relaxing. The depression scale assesses hopelessness, distress, self-deprecation and life-deprecation. The anxiety scale assesses anxiety as a situation but also as a subjective experience that has an impact on the individual’s daily life. 

The positive and negative effects questionnaire (PANAS) aims at evaluating positive and negative situations. The positive state includes the feelings of joy, excitement, alertness and activity. The negative attitude is related to the discomfort that the individual feels in relation to the feelings of fear, anger and guilt. The questionnaire consists of 20 questions, which are divided into ten questions concerning positive effects and ten concerning negative effects. The 4-point Likert scale was used for the measurement, where 1 = not at all and 4 = extremely [13], including mentions of specific feelings such as joy, excitement, alertness, activity, fear, anger and guilt. 

Furthermore, the BRIEFCOPE questionnaire is a stress assessment questionnaire, which is used to describe stress coping strategies in a given time period or in a specific situation. The questions are answered on a 5-Likert scale, where 1 means “I don’t act this way at all” and 4 means “I very often act this way” [14]. 

The Brief Resilience Scale (BRS) [15] aims to evaluate the individual’s ability to cope with the difficulties and stressful situations he experiences in his everyday life. It includes 6 questions, which are answered on a 5-point Likert scale from 0 = totally disagree to 5 = totally agree.

Finally, the Life Orientation Test (LOT) [16] is a questionnaire used to assess levels of optimism and pessimism. The Lot consists of 10 questions, which are divided into 3 questions where positive elements are described, 3 questions where negative elements are described and 4 elements that are not scored. The answers are evaluated on a 4-point Likert scale and the score ranges from 0 “strongly disagree” to 4 “strongly agree”. All of the above research questionnaires were used after license agreement from original authors. 

### 2.3. Statistical Analysis

For the analysis of the quantitative and qualitative variables, different statistical characteristics were used, such as the mean (standard deviation), the median (interquartile range) and the absolute and relative frequencies. Quantitative variables were tested for normality using the Kolmogorov–Smirnov criterion. Spearman’s coefficient was aimed at the correlation of two variables. For the comparison of variables between two groups, the Student’s *t*-test or the Mann–Whitney test was used. Analysis of variance (ANOVA) or the Kruskal–Wallis test was used to compare continuous variables across more than two groups. The Bonferroni correction was used to detect a type 1 error due to continuous comparisons. The DASS-21 subscales were used for multiple linear regression analysis. The regression equation included the subscales BriefCOPE, BRS, LOT and PANAS, demographics, education information and factors related to COVID-19 measures. Linear regression analyses were used to identify the effects of the adjusted regression coefficients (β) with standard errors. A linear regression analysis was performed to find the dependent variables of logarithmic transformations. Cronbach’s alpha factor was used to assess the internal reliability of the questionnaire. Analysis was performed with SPSS statistical software (version 22.0) and statistical significance was set at *p* < 0.05.

## 3. Results

The sample included 288 students (84.4% females), whose demographic data are presented in Table 1. Most of the students (58.0%) were 18–22 years old and 45.5% lived in their family home. A total of 91.0% of the population had siblings and 61.5% were living with their family. Almost all participants (92.0%) were studying in a health-related department and 98.6% were undergraduate students. Most students (76.7%) were attending a 4-year school program and 29.2% were in the second year. Moreover, 30.9% of the participants had 5–8 h online education per week. Moreover, 13.2% of them had mental support in the university for pressure management due to the application of new measures in relation to the COVID-19 pandemic and 19.1% mentioned support from a professor. In addition, 4.9% knew someone infected by COVID-19 and only one of the students had been infected. Furthermore, 95.1% were informed about measures for COVID-19 prevention and 92% were applying those measures in their everyday life and household.

Mean values of BRIEFCOPE, LOT, Resilience and PANAS scales are presented in Table 2. Additionally, in Table 3 DASS-21 subscales are described. Depression was experienced by 44.8% of the sample. Anxiety was experienced by 36.8% of the students and stress was mentioned by 40.3% of the total study population (Figure 1).

Female students had significantly greater anxiety and stress, compared to males (Table 4). Moreover, they used Eliciting Supportive Actions from others significantly more than males. Resilience score was significantly greater in males, as it was for the Positive Affect Score. On the contrary, the negative Affect Score was significantly greater in females.

Students’ scores in all under study scales are presented in Table 5 by years of study. Only the active/positive coping score differed significantly among year of study. More specifically, after Bonferroni correction it was found that students in the fourth year were using significantly more active/positive coping strategies than students in the first (*p* = 0.016) or second year of study (*p* = 0.005). All other scores were similar across all years of study.

Depression and stress scales were significantly positively correlated with Behavioral withdrawing, Drug use, Avoidance coping and Negative feelings (Table 6). Moreover, the depression scale was significantly negatively associated with the religion subscale. Anxiety scale was significantly and positively correlated with Behavioral withdrawing, Drug use, Eliciting supportive actions from others, Avoidance coping and Negative feelings. On the contrary, stress, anxiety and depression scales were significantly negatively correlated with active/positive coping, LOT and resilience scales and positively correlated with Negative Affect Score. Depression and anxiety scales were significantly negatively correlated with a positive Affect Score.

Multiple linear regression analysis was conducted, and it was reported that older students, participants who experienced more behavioral withdrawing and Avoidance coping as coping strategies, participants with a higher negative affect score, participants with a lower positive affect score and those with greater optimism had significantly greater depression signs (Table 7). Moreover, students using more Drug use, Negative feelings as coping strategies, participants with greater negative affect score as well as participants with lower resilience had significantly greater anxiety and stress expressions. 

## 4. Discussion

The object of this study was to evaluate anxiety, stress, depression as well as their associated variables among university students in Greece during the COVID-19 pandemic. The data were congregated during the first wave of the COVID-19 pandemic, including the time period between June and August 2020. Almost half of the sample reported depression (severe and extremely severe depression), one third reported anxiety (severe and extremely severe anxiety) and 1 out of 4 reported stress (severe and extremely severe stress). The mean (SD) value of depression, anxiety and stress for all students was found to be normal, respectively. The median (IQR) values of their subscales were 4 (1–9) for depression, 2 (0–5) for anxiety and 6 (3–11) for stress. 

The present study showed higher mean values of DASS than those in the study of Deng et al. [17] in college students in Wuhan, and lower mean values compared to university students in Egypt [18], which were found at the mild level. However, in our study the percentage of students who reported severe and extreme severe depression, anxiety and stress was much higher. Moreover, the prevalence of at least mild depression, anxiety and stress and the incidence of severe depression, anxiety and stress in university students was higher in comparison with our previous study in Greek nurses [19]. 

A study in Bangladeshi students [20] revealed that mild to severe stress was reported by one out of four students. Respectively, this percentage was for one third of the sample for anxiety and almost half of the participants for depression. Additionally, the study suggested that stress among university students had a greater effect on their psychology than in college students. Many studies have indicated that students report steadily more mental health issues than the general population [21,22,23]. There are multiple stress factors, such as life-stage transitions, study load, academic pressure during exam periods, intense pressure for academic success, problems associated with their accommodation, acclimation to new social and geographical environments and concerns about the future [24,25]. During the COVID-19 pandemic, the adoption of protective measures, such as the closure of schools and universities, caused disruptions of daily life and lifetime stressors [18]. Higher stress in university students may be due to hampered educational activity during this period and could be dealt with through necessary arrangements for online classes. A stress factor for young people can be incertitude related to progression in academic life [26]. 

The current study suggested that anxiety, stress and depression were more common in female students. Findings in earlier studies agree with that result [27,28,29,30]. So far, it has been suggested that the differences in mental health problems between genders are influenced by several factors associated with the environment and the genetic and physiological background. [31,32]. Moreover, females used Eliciting supportive actions from others significantly more than males, and they had a greater negative affect score. Males were more resilient with a greater positive affect score. The same results concluded the study of Adjepong et al. [33], in which female students demonstrated lower resilience scores and higher negative mood scores as well as they reported increased stress levels during the pandemic. This finding is consistent with previous work indicating that low resilience is associated with a decreased capability to deal with stress [34]. In contrast, males have been reported to have greater resilience during adolescence and early adulthood than females, but differences disappear in older adulthood [35]. 

Students in the fourth year of study used significantly more active/positive coping strategy than students in the first or second year. Previous studies have defined students’ characteristics, such as age and year of study, as significant prediction factors stress and coping [36,37,38,39]. A study in Spanish nursing students found that nursing students’ year of study greatly affected their overall stress experiences, with junior nursing students having higher stress perceptions compared to senior nursing students. Moreover, age was predicted to be related with a higher utilization of coping skills in nursing students, with those that were older able to apply coping skills more effectively than those who were younger. In another study [27] in Ethiopian university students, participants in their first and second year were more stressed. A study by Wang et al. [40] regarding the general population of China during the COVID epidemic reported a higher psychological impact on respondents aged 12–21.4 years. 

Moreover, in the present study it was suggested that those who had greater depression were older students, participants using more coping strategies behavioral withdrawing and Avoidance coping, participants with a higher negative affect score, participants with lower positive affect score and those with greater optimism. Those using more substance use, Negative feelings as strategies to cope with stress, participants with greater negative affect score as well as those with lower resilience had significantly greater anxiety and stress. Association between stress and coping is consistent with previous studies [39,41]. The occurrence of stress factors and Avoidance coping as a coping mechanism predicted the presence of psychopathological symptomatology, while an active coping mechanism predicted a greater satisfaction with life by dealing with such stress factors [42]. Farrer et al. [43] reported that the risk of suffering from major depressions was significantly higher for Australian students in their first year of study. This leads to conclude that being in confinement was different in comparison to other stress factors, such as the anticipatory anxiety when writing exams and perhaps requires coping strategies that differ from those for exam preparation.

## 5. Strengths and Limitations of the Study

This study had the major strength that students from different Faculties of Universities in Greece, were enrolled. However, the period of the study instruments’ completion was quite limited, during the latest period of the first wave of COVID-19 pandemic (June–August 2020). Additionally, due to having a cross-sectional design, it was difficult to determine whether any of the psychological impacts had already occurred or had recently developed. Considering the fact that the variables examined were dynamic, the use of a longitudinal study may be useful in order to track its improvement and/or aggravation. Second, we used an online survey which may have contributed to non-response bias in the study results. Furthermore, the enrollment of more participants leading to larger samples may extract more generalized findings of great interest. 

## 6. Conclusions

Many students were affected by anxiety, stress, and depression. Several factors such as gender, year of study, age, positive and negative affect score, life orientation test score and coping strategies were identified as variables contributing to either of the common mental health problems. Special consideration must be given to the most affected groups such as female students. Thus, the increasing need for better surveillance of students’ mental wellbeing and subsequent counseling is even more evident now.

## Figures and Tables

**Figure 1 healthcare-10-02573-f001:**
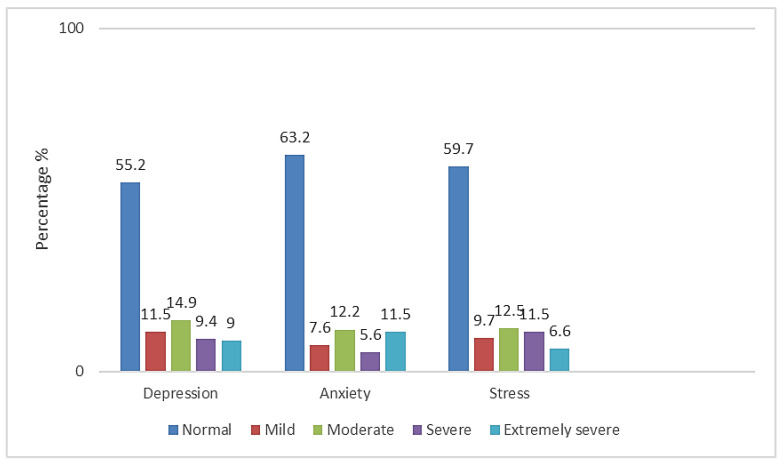
Measured levels of depression, anxiety and stress.

**Table 1 healthcare-10-02573-t001:** Participants’ characteristics.

	N	%
**Gender**		
Female	243	84.4
Male	45	15.6
**Age (years)**		
18–22	167	58.0
23–35	70	24.3
36–45	31	10.8
>45	20	6.9
**Place of residence**		
Family home	131	45.5
Alone in rental place	97	33.7
Alone in own place	50	17.4
Dorms	10	3.5
**Siblings**	262	91.0
**Living in the same house with:**		
No one	79	27.4
Partner	32	11.1
Family	177	61.5
**Health related department**	265	92.0
**Student**		
Undergraduate	284	98.6
Postgraduate	4	1.4
**Year of study**		
1st	59	20.5
2nd	84	29.2
3rd	54	18.8
4th	26	9.0
At least 5th	65	22.6
**Duration of school for graduation (years)**		
2	17	5.9
4	221	76.7
5	49	17.0
6	1	0.3
**Hours of online education per week**		
2–4	81	28.1
5–8	89	30.9
9–12	76	26.4
>13	42	14.6
Mental support received in university for pressure management due to the application of new measures regarding the COVID-19 pandemic	38	13.2
Mental support received from professors for pressure management due to the application of new measures regarding the COVID-19 pandemic	55	19.1
Applying measures for preventing the spread of COVID-19 in the household	265	92.0
Knowing someone infected by SARS-CoV2	14	4.9
**If yes, define**		
Working environment member	4	1.4
Family member	4	1.4
Social acquaintance	3	1.0
Friend	6	2.1
**Daily hours of sleep**		
<7 h	75	26.0
7–9 h	201	69.8
>10 h	12	4.2
**Been infected by COVID-19**	1	0,3
**If yes, did you stay in quarantine**		
Όχι	0	0.0
1–2 weeks	1	100.0
>4 weeks	0	0.0
Attend webinar for COVID19 and ways of protection against it	176	61.1
Informed about measures for COVID-19 prevention	274	95.1
Applying measures towards preventing the spread of COVID-19 in everyday life	265	92.0

**Table 2 healthcare-10-02573-t002:** Description of BRIEFCOPE, LOT, Resilience and PANAS scales.

	Minimum	Maximum	Mean (SD)	Cronbach’s a
**BRIEFCOPE subscales**				
Positive/active coping	10.0	36.0	26.4 (4.8)	0.76
Behavioral withdrawing	3.0	12.0	4.8 (1.9)	0.73
Drug use	2.0	8.0	2.4 (1.0)	0.94
Eliciting supportive actions from others	4.0	16.0	10.2 (3.4)	0.90
Religion	2.0	8.0	4.2 (2.0)	0.79
Humor	2.0	8.0	4.7 (1.7)	0.73
Avoidance coping	3.0	12.0	7.9 (1.9)	0.70
Expression of negative feeilings	3.0	12.0	7.2 (2.1)	0.71
**Life Orientation Test score**	0.0	24.0	13.7 (5.5)	0.74
**Resilience score**	6.0	30.0	19.5 (4.6)	0.86
**PANAS**				
Positive Affect Score	12.0	44.0	26.2 (6.3)	0.76
Negative Affect Score	10.0	45.0	22.3 (7.3)	0.85

**Table 3 healthcare-10-02573-t003:** Description of DASS-21 subscales.

	Minimum	Maximum	Mean (SD)	Median (IQR)	Cronbach’s a
Depression	0.0	21.0	5.3 (5.1)	4 (1–9)	0.86
Anxiety	0.0	21.0	3.6 (4.5)	2 (0–5)	0.87
Stress	0.0	21.0	7.0 (5.4)	6 (3–11)	0.89

**Table 4 healthcare-10-02573-t004:** Students’ scores by gender.

	Gender		
	Males	Females	
	*Median (IQR)*	*Median (IQR)*	*p* *Mann-Whitney test*
Depression	2 (0–6)	4 (1–9)	0.035
Anxiety	0 (0–3)	2 (0–6)	<0.001
Stress	4 (1–8)	6 (3–11)	<0.001
	*Mean (SD)*	*Mean (SD)*	*p* *Student’s t-test*
BRIEFCOPE subscales			
Positive/active coping	26.6 (5)	26.3 (4.8)	0.658
Behavioral withdrawing	4.6 (1.8)	4.8 (1.9)	0.585
Drug use	2.5 (1.3)	2.3 (1)	0.313
Eliciting supportive actions from others	9.1 (3.6)	10.4 (3.3)	0.020
Religion	3.8 (1.9)	4.3 (2)	0.115
Humor	4.9 (1.7)	4.7 (1.7)	0.358
Avoidance coping	7.4 (2.2)	7.9 (1.9)	0.124
Expression of negative feelings	6.8 (2.3)	7.3 (2)	0.092
Life Orientation Test score	13.2 (5.8)	13.8 (5.4)	0.541
Resilience score	20.8 (5)	19.3 (4.5)	0.039
PANAS			
Positive Affect Score	29 (7.5)	25.7 (6)	0.001
Negative Affect Score	18.4 (6.3)	23 (7.3)	<0.001

**Table 5 healthcare-10-02573-t005:** Students’ scores by year of study.

	Year of Study					
	1st	2nd	3rd	4th	5th	
	*Median (IQR)*	*Median (IQR)*	*Median (IQR)*	*Median (IQR)*	*Median (IQR)*	*p Kruskal-Wallis* *test*
**Depression**	5 (2–9)	3 (1–8)	4 (1–7)	5 (1–10)	4 (1–10)	0.466
**Anxiety**	2 (0–5)	1.5 (0–5)	1.5 (0–5)	2 (0–8)	2 (0–5)	0.987
**Stress**	6 (3–11)	4 (1–11)	6 (3–9)	6 (4–14)	6 (3–11)	0.259
	*Mean (SD)*	*Mean (SD)*	*Mean (SD)*	*Mean (SD)*	*Mean (SD)*	*p ANOVA*
**BRIEFCOPE subscales**						
Positive/active coping	25.8 (4.5)	25.7 (4.9)	26.6 (4.8)	29.4 (3.6)	26.3 (4.9)	0.010
Behavioral withdrawing	4.7 (2)	4.7 (2)	4.6 (1.7)	4.7 (1.6)	5.1 (1.9)	0.542
Drug use	2.3 (1)	2.4 (1.1)	2.2 (0.6)	2.3 (1.2)	2.5 (1.2)	0.459
Eliciting supportive actions from others	10.3 (3.4)	10.2 (3.5)	10.3 (3.4)	10.1 (3.3)	10 (3.4)	0.981
Religion	4.1 (1.9)	4 (2)	4.1 (1.7)	5 (2.1)	4.3 (2.1)	0.281
Humor	4.5 (1.7)	4.5 (1.7)	4.9 (1.6)	5.4 (1.5)	4.8 (1.7)	0.094
Avoidance coping	7.7 (1.9)	7.6 (2)	7.8 (1.5)	8.3 (2.3)	8.2 (1.9)	0.190
Expression of negative feelings	7 (1.8)	7.2 (2.2)	7.5 (2)	7.7 (2.3)	7.1 (2.1)	0.496
**Life Orientation Test score**	13.8 (5.3)	13.8 (5.4)	14.2 (5.9)	13.6 (5.6)	13.2 (5.4)	0.892
**Resilience score**	19 (4.1)	19.1 (4.9)	20.4 (4.1)	19.1 (4.6)	20 (4.9)	0.329
**PANAS**						
**Positive Affect Score**	24.9 (5.6)	25.4 (6.8)	27.7 (6.3)	26.2 (5.2)	27.2 (6.5)	0.064
**Negative Affect Score:**	23.3 (7.3)	20.9 (7.5)	22.2 (6.2)	23 (8.1)	22.7 (7.5)	0.331

**Table 6 healthcare-10-02573-t006:** Spearman’s correlation coefficients of DASS-21 with BRIEFCOPE, LOT, RES and PANAS.

	Depression	Anxiety	Stress
Positive/active coping	−0.25 ***	−0.15 **	−0.14 *
Behavioral withdrawing	0.47 ***	0.37 ***	0.40 ***
Drug use	0.13 *	0.15 *	0.12 *
Eliciting supportive actions from others	0.03	0.14 *	0.11
Religion	−0.18 **	−0.02	−0.06
Humor	0.07	0.04	0.07
Avoidance coping	0.28 ***	0.20 ***	0.26 ***
Negative feelings	0.22 ***	0.26 ***	0.26 ***
Life Orientation Test score	−0.48 ***	−0.35 ***	−0.37 ***
Resilience score	−0.45 ***	−0.43 ***	−0.39 ***
Positive Affect Score	−0.25 ***	−0.12 *	−0.10
Negative Affect Score	0.58 ***	0.55 ***	0.65 ***

* *p* < 0.05; ** *p* < 0.01; *** *p* < 0.001.

**Table 7 healthcare-10-02573-t007:** Multiple linear regression analysis with DASS-21 as the dependent variable and students’ characteristics, BRIEFCOPE, LOT, RES and PANAS as independents, using the stepwise method.

	β +	SE ++	95% CI	*p*
*Depression*				
Age	−0.056	0.019	−0.093; −0.019	0.003
Behavioral withdrawing	0.031	0.011	0.011; 0.052	0.003
Avoidance coping	0.038	0.009	0.021; 0.056	<0.001
LOT score	−0.016	0.004	−0.023; −0.009	<0.001
Positive Affect Score	−0.006	0.003	−0.012; −0.001	0.029
Negative Affect Score	0.024	0.003	0.019; 0.029	<0.001
*Anxiety*				
Drug use	0.041	0.018	0.006; 0.076	0.023
Negative feelings	0.021	0.009	0.003; 0.039	0.021
Resilience	−0.021	0.004	−0.030; −0.013	<0.001
Negative Affect Score	0.026	0.003	0.021; 0.032	<0.001
*Stress*				
Negative feelings	0.021	0.009	0.003; 0.039	0.021
Drug use	0.041	0.018	0.006; 0.076	0.023
Resilience	−0.021	0.004	−0.030; −0.013	<0.001
Negative Affect Score	0.026	0.003	0.021; 0.032	<0.001

+ regression coefficient; ++ Standard Error.

## Data Availability

All the data generated during this study are included in this published article.

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
