# Peer review of "Anxiety, Stress and the Resilience of University Students during the First Wave of the COVID-19 Pandemic"

_healthcare, 2022, doi:10.3390/healthcare10122573_

Round 1

Reviewer 1 Report

Comments to the Author

The manuscript titled "Anxiety, stress and resilience of University Students during the first wave of COVID-19 pandemic" analyzes the psychological difficulties experienced by university students in Greece during the first wave of the outbreak. However, there are several points that require further clarity;

1- Pages 1-9: If the study population does not consist of COVID-19 survivors, we can talk about their depression, anxiety, or stress levels, but these cannot be considered a SYMPTOM! please revise throughout the text.

2- Page 2, Line 70: You must indicate in which academic year the study was carried out.

You should also indicate which faculty students participated in the study. If there is more than one faculty, make a statistical comparison (between faculties).

Was there any chronic disease (COPD, Asthma, etc.) in the population?

3- Page 2, Lines 75-76: What is the Ethic confirmation number?

4- Page 2, Line 79: This section is very limited. Provide more details to demonstrate repeatability of all your scales (DASS, PANAS, BRS, and LOT).

5- Page 2, Line 93: Have the data been tested for normality? If yes, please specify which method was used.

6- Page 3, Lines 98-99: Revise the ANOVA sentence to "groups of more than two".

7- Page 3, Line 103: Revise from ''covid-19'' to ''COVID-19'' or ''SARS-CoV-2'' in all text.

8- Page 3, Line 122: You say university students, but half of the population is over 23 (Table 1)? Please explain.

9- Page 6, Line 143: The Wilcoxon test is used to compare samples taken from the same subject at two different times and is a non-parametric test. Please check your method.! I also noticed that you didn't mention this test in the statistics section.

10- Page 8, Lines 169-185: You should move numeric data away from this section. In this section, discuss only the major findings of the study.

11- Page 9, Lines 240-243: Is it really the strength of this work that research has been done in the first wave of COVID-19? Hasn't it been too long and too many similar studies have been published so far? please explain.

GENERAL COMMENTS:

1. The manuscript requires language improvement.

2. The discussion section should be improved significantly. Literature review is nonadequacy.

3. Abstract should be re-edited after changes made in the article.

Author Response

√ 1- Pages 1-9: If the study population does not consist of COVID-19 survivors, we can talk about their depression, anxiety, or stress levels, but these cannot be considered a SYMPTOM! please revise throughout the text.

ANSWER: No comment

√2- Page 2, Line 70: You must indicate in which academic year the study was carried out.

You should also indicate which faculty students participated in the study. If there is more than one faculty, make a statistical comparison (between faculties).

Was there any chronic disease (COPD, Asthma, etc.) in the population?

ANSWER: The study was carried out the academic year 2019-2020 (the sample was between 15/06/2020 and 18/8/2020). The majority of the participants were nursing students.

Five Greek Universities were participated in this study.

No chronic disease report was used in the research.

√3- Page 2, Lines 75-76: What is the Ethic confirmation number?

ANSWER: It is mentioned above the references list.

√4- Page 2, Line 79: This section is very limited. Provide more details to demonstrate repeatability of all your scales (DASS, PANAS, BRS, and LOT).

ANSWER: Confirmed

√5- Page 2, Line 93: Have the data been tested for normality? If yes, please specify which method was used.

ANSWER: The Kolmogorov-Smirnov test was used for testing normality. It was added in the "Statistical analysis" section

√6- Page 3, Lines 98-99: Revise the ANOVA sentence to "groups of more than two".

ANSWER: The sentence was corrected.

√7- Page 3, Line 103: Revise from ''covid-19'' to ''COVID-19'' or ''SARS-CoV-2'' in all text.

ANSWER: Confirmed

√8- Page 3, Line 122: You say university students, but half of the population is over 23 (Table 1)? Please explain. 

ANSWER: 121 (42%) out of students are above 23 years old. In Greece, educational committees accept students who can study at universities despite their age.

√9- Page 6, Line 143: The Wilcoxon test is used to compare samples taken from the same subject at two different times and is a non-parametric test. Please check your method.! I also noticed that you didn't mention this test in the statistics section.

ANSWER: Wilcoxon test was written by mistake, although Kruskal-Wallis had been conducted. It was corrected. 

10- Page 8, Lines 169-185: You should move numeric data away from this section. In this section, discuss only the major

findings of the study.

ANSWER: Confirmed.

11- Page 9, Lines 240-243: Is it really the strength of this work that research has been done in the first wave of COVID-19? Hasn't it been too long and too many similar studies have been published so far? please explain.

ANSWER: Corrected

GENERAL COMMENTS:

  1. The manuscript requires language improvement. Confirmed
  2. The discussion section should be improved significantly. Literature review is nonadequacy. Confirmed
  3. Abstract should be re-edited after changes made in the article. Confirmed

Reviewer 2 Report

The authors focused on “Anxiety, stress and resilience of University Students during the 2 first wave of COVID-19 pandemic “. In the reviewer's opinion, the manuscript in its current form cannot be recommended for publication. The authors are asked to take into consideration the comments of the reviewer listed below and respond to them:

  • Line 7-9 : It is recommended to keep only one contact email address. Therefore, it is necessary to remove all email addresses other than correspondence one.

  • Line 19-20 (quote) “Methods: 288 university students completed a special questionnaire after consent “. Please explain what "a special questionnaire" means, or it is recommended to paraphrase the quote and provide the name of the “a special questionnaire” used.

  • Line 71-72 ( quote) “A cross-sectional study was conducted in Greek Universities on the stress of students during the first period of the Covid-19 pandemic “. There is no information on the dates of the first period of the Covid-19 pandemic considered by the authors. Dates are required.

  • Line 80-81 (quote) “The first data recorded by the participants were demographic characteristics related 80 to age, sex, place of residence, religion, and educational level. “. Please explain the reasons for collecting religious data if such data is not addressed in the discussion paragraph.

  • Unfortunately,  were noted discrepancies between data information in Line 71- 72 [ (quote ) “A cross-sectional study was conducted in Greek Universities on the stress of students during the first period of the Covid-19 pandemic “] and Line 168-169 [ (quote) “ The data were congregated during the first and the second wave of COVID-19 168 pandemic, including time period between June and December 2020. “ ] .

The quoted data information requires urgent clarification. Thus, please explain: Were the data congregated during the first and the second wave of COVID-19 168 pandemic, including time period between June and December 2020 or ( having in mind information in line 71- 72 ) during the first period of the Covid-19 pandemic ONLY ?

  • It is recommended to establish “Strengths and Limitations “ paragraph.

  • List of References do not comply with the journal guidelines. Therefore all references must be revised and improved.

Author Response

The authors focused on “Anxiety, stress and resilience of University Students during the 2 first wave of COVID-19 pandemic.

“In the reviewer's opinion, the manuscript in its current form cannot be recommended for publication. The authors are asked to take into consideration the comments of the reviewer listed below and respond to them:

  • Line 7-9: It is recommended to keep only one contact email address. Therefore, it is necessary to remove all email addresses other than correspondence one.

ANSWER: The automatic system of mdpi platform completed the email addresses within the article.

  • Line 19-20 (quote) “Methods: 288 university students completed a special questionnaire after consent “. Please explain what "a special questionnaire" means, or it is recommended to paraphrase the quote and provide the name of the “a special questionnaire” used.

ANSWER: Confirmed

  • Line 71-72 (quote) “A cross-sectional study was conducted in Greek Universities on the stress of students during the first period of the Covid-19 pandemic “. There is no information on the dates of the first period of the Covid19 pandemic considered by the authors. Dates are required.

ANSWER: Confirmed

  • Line 80-81 (quote) “The first data recorded by the participants were demographic characteristics related 80 to age, sex, place of residence, religion, and educational level. “. Please explain the reasons for collecting religious data if such data is not addressed in the discussion paragraph.

ANSWER: Corrected

  • Unfortunately, were noted discrepancies between data information in Line 71- 72 [ (quote ) “A cross-sectional study was conducted in Greek Universities on the stress of students during the first period of the Covid-19 pandemic “] and Line 168-169 [ (quote) “ The data were congregated during the first and the second wave of COVID-19 168 pandemic, including time period between June and December 2020. “ ] .The quoted data information requires urgent clarification. Thus, please explain: Were the data congregated during the first and the second wave of COVID-19 168 pandemic, including time period between June and December 2020

or ( having in mind information in line 71- 72 ) during the first period of the Covid-19 pandemic ONLY ?

ANSWER: Corrected

  • It is recommended to establish “Strengths and Limitations “ paragraph.

ANSWER: Confirmed

  • List of References do not comply with the journal guidelines. Therefore all references must be revised and improved

ANSWER: Checked

Round 2

Reviewer 1 Report

The authors answered to my comments properly, improving the quality of their manuscript. However, I still have a concern about the use of the word ''symptoms''. Please read the study below and decide whether appropriate of this word is for your study.

https://doi.org/10.1007/s11019-013-9501-5

Sincerely.

Author Response

R1 comments:

The authors answered to my comments properly, improving the quality of their manuscript. However, I still have a concern about the use of the word ''symptoms''. Please read the study below and decide whether appropriate of this word is for your study.

https://doi.org/10.1007/s11019-013-9501-5

Sincerely.

ANSWER: According to the definition of Eriksen, ‘’What is called symptom?’’, we have decided to remove the word symptom because the meaning of the sentence is not altered after removal, though. Thank you!

Reviewer 2 Report

Although the authors referred in good manner to most of the reviewer's comments and recommendations, and revised / improved manuscript; however it unfortunately happens that the list of references still does not meet the requirements of the journal. Thus, please take into consideration reviewer's hint: Name initials should be followed by semicolons, not commas between authors data.

Author Response

R2 comments:

Although the authors referred in good manner to most of the reviewer's comments and recommendations, and revised / improved manuscript; however it unfortunately happens that the list of references still does not meet the requirements of the journal. Thus, please take into consideration reviewer's hint: Name initials should be followed by semicolons, not commas between authors data.

ANSWER: CHECKED, OK.